# Uterine Fibroids and Infertility

**DOI:** 10.3390/diagnostics11081455

**Published:** 2021-08-12

**Authors:** Damaris Freytag, Veronika Günther, Nicolai Maass, Ibrahim Alkatout

**Affiliations:** Department of Obstetrics and Gynecology, University Hospital of Schleswig-Holstein, Campus Kiel, Arnold-Heller-Strasse 3, 24105 Kiel, Germany; veronika.guenther@uksh.de (V.G.); Nicolai.maass@uksh.de (N.M.); ibrahim.alkatout@uksh.de (I.A.)

**Keywords:** infertility, uterine fibroids, pregnancy rates, implantation, hysteroscopy, laparoscopy

## Abstract

Infertility is a disease of the reproductive system defined by the failure to achieve a clinical pregnancy after 12 months or more of regular unprotected sexual intercourse. Uterine fibroids are the most common tumor in women, and their prevalence is high in patients with infertility. Fibroids may be the sole cause of infertility in 2–3% of women. Depending on their location in the uterus, fibroids have been implicated in recurrent pregnancy loss as well as infertility. Pregnancy and live birth rates appear to be low in women with submucosal fibroids; their resection has been shown to improve pregnancy rates. In contrast, subserosal fibroids do not affect fertility outcomes and their removal does not confer any benefit. Intramural fibroids appear to reduce fertility, but recommendations concerning their treatment remain unclear. Myomectomy should be discussed individually with the patient; other potential symptoms such as dysmenorrhea or bleeding disorders should be included in the indication for surgery.

## 1. Introduction

Infertility is an important social and economic problem because many couples plan their families much later in life now than couples did three decades ago. With increasing age, women have fewer chances of natural fertilization and may be less likely to maintain a pregnancy. Consequently, many couples need assisted reproductive technology (ART). However, a large number of women undergoing in vitro fertilization (IVF) suffer from infertility in the form of recurrent implantation failure [1].

Infertility has been diversely defined from clinical, demographic, and epidemiological viewpoints. It has also been viewed as a disability. By clinical definition, infertility is a disease of the reproductive system defined by the failure to achieve a clinical pregnancy after 12 months or more of regular unprotected sexual intercourse [2].

Human reproduction is an inefficient process, because only about 30% of conceptions result in a live birth. Although exact percentages are impossible to access, it has been estimated that approximately 30% of embryos are lost at the preimplantation stage, while 30% are lost after implantation in the uterus and only detected by a positive serum human chorionic gonadotropin (hCG) test in the absence of ultrasound findings. Ten percent are clinical miscarriages, including abortion and stillbirth (Figure 1) [3]. Most pregnancy wastage is caused by the quality of the embryos themselves. In about 70% of cases, significant chromosome abnormalities are responsible for sporadic abortions. The problem of early abortion became known in the era of IVF treatment, because the exact date of embryo transfer and expected implantation can be predicted in IVF treatment. Hence, recurrent implantation failure became a clinically identifiable phenomenon.

The development of a pregnancy is a multifaceted process. It can be influenced and hindered by various systemic and local factors, such as maternal age, oocyte and sperm quality, parental chromosomal abnormalities, genetic or metabolic abnormalities of the embryo, poor uterine receptivity, and immunological imbalance at the implantation site. Gynecological conditions that could influence implantation rates include endometriosis, uterine fibroids, hydrosalpinges, and endometrial polyps. Finally, factors such as lifestyle, smoking, alcohol, drugs, and obesity causing insulin resistance might impair the success of reproduction [4,5,6].

In the following, we discuss uterine fibroids as a possible cause of infertility, their investigation, and treatment options.

Uterine fibroids, also known as uterine leiomyomas or fibroids, are benign smooth muscle tumors of the uterus that affect women of reproductive age. Fibroids have both smooth muscle and fibroblast components, in addition to a substantial fibrous extracellular matrix, all of which contribute to the process of pathogenesis. Fibroids are extremely heterogeneous in terms of pathophysiology, size, location, and clinical symptoms [7]. While some women have no symptoms, others experience dysmenorrhea or hypermenorrhea. The symptoms and their severity may differ, depending on the size and location of the fibroids. The most common presenting symptom is heavy menstrual bleeding, which may lead to anemia, fatigue, or painful periods. Other possible symptoms include lower back pain, pelvic pressure or pain, and pain during intercourse. In the presence of fibroids beyond a certain size, pressure on the bladder or bowel may result in increased micturition frequency or retention, pain, or constipation. Uterine fibroids may also be associated with reproductive problems such as infertility, recurrent pregnancy loss, and adverse obstetric outcomes [8,9,10].

Although the cause of uterine fibroids is largely unknown, genetic and epigenetic factors influence the risk of development of uterine fibroids, like age, race and ethnicity, family history, body mass index, early-life environmental exposure to toxins, or vitamin D deficiency [11,12]. Vitamin D deficiency is characterized by a serum vitamin D level of <20 ng/mL, with the normal level being ≥30 ng/mL. Low levels of serum vitamin D are associated with increased sizes of uterine fibroids in different ethnic groups [13]. African–American women not only have a higher rate of fibroid disease but also are more likely to be vitamin D deficient compared to women of other groups [14]. In detail, the expression of the vitamin D receptor seems to be lower in leiomyomas than in nonneoplastic myometrial tissue [15]. Although the mechanisms by which vitamin D exerts its effects in fibroids are conveyed through the regulation of gene expression, some of these effects are also mediated by the modulation of intracellular signaling pathways, thus suggesting that vitamin D is directly or indirectly connected to multiple cellular processes [12].

Traditionally, fibroids are classified by their location in the uterus. They may be divided into cervical, submucosal, subserosal, and intramural fibroids. The International Federation of Gynecology and Obstetrics (FIGO) uses the following classification system, as shown in Figure 2 and described in Table 1.

## 2. Uterine Fibroids and Infertility

Fibroids may be the sole cause of infertility in 2–3% of women [16,17]. Depending on their location in the uterus, fibroids have been implicated in recurrent pregnancy loss as well as infertility.

Implantation has a well-defined starting point and then proceeds rather slowly for several weeks; the time of its conclusion cannot be predicted in advance. Clinically, implantation is considered to be successful when there is ultrasonic evidence of an intrauterine gestational sac, which usually forms at about five weeks of gestation. In contrast, implantation failure is defined as the absence of an intrauterine gestational sac on ultrasound. Implantation failure may occur in the rather early stages of attachment or migration. The absence of objective evidence of pregnancy is a negative hCG test. Implantation failure may also occur later on, after successful migration of the embryo through the luminal surface of the endometrium. HCG, which is produced by the embryo, can be detected in a blood or urine test. However, the process may be disrupted before the emergence of an intrauterine gestational sac; this condition is known as a biochemical pregnancy [18].

An evaluation of outcomes in women with infertility revealed that those with fibroids in any location had significantly lower rates of clinical pregnancy, implantation, ongoing pregnancy, and live birth rates compared to controls. In addition, the spontaneous abortion rate was significantly higher in women with fibroids. No difference was noted in regard of preterm delivery rates [19].

In the following, fibroids are divided according to their location and their impact on fertility:

Women with subserosal fibroids did not differ from those without fibroids with regard to implantation rates, clinical pregnancy rates, live birth rates, and abortion rates. Thus, subserosal fibroids do not seem to affect fertility [19]. In contrast, submucosal and intramural fibroids that distort the endometrial cavity are associated with lower pregnancy, implantation, and delivery rates in women undergoing in vitro fertilization (IVF) compared to infertile women without fibroids [20,21,22]. Furthermore, there is a higher risk of infertility when the endometrial cavity is distorted by submucosal fibroids [23,24]. Pregnancy and delivery rates appear to be improved after resection of submucosal fibroids, especially when fibroids are the sole identifiable cause of infertility [21,24,25]. The exact pathomechanism as to how intramural fibroids affect the overlying endometrium and influence receptivity is not fully understood. Fibroids may affect implantation by several mechanisms, including increased uterine contractility, deranged cytokine profile, abnormal vascularization, and chronic inflammation [26]. In the following, we will address the pathophysiology of intramural fibroids.

## 3. Pathophysiology

HOXA 10 is a homeobox-containing transcription factor that is essential for embryonic uterine development as well as proper adult endometrial development during each menstrual cycle [27]. HOXA 10 expression is necessary for endometrial receptivity [28,29,30]. Glycodelin is a secretory glycoprotein that affects cell proliferation, differentiation, adhesion, and motility [31]. Glycodelin is responsible for promoting angiogenesis and suppressing natural killer cells during implantation. Normally, HOXA 10 and glycodelin are reduced during the follicular phase and increased during implantation. In cases of intramural fibroids, both HOXA 10 and glycodelin are reduced during implantation, which may lead to embryo implantation failure and cause infertility [30].

The uterine junctional zone is the inner third of the myometrium and the layer that immediately abuts the endometrium. The layer differs architecturally from the rest of the myometrium and appears to be the origin of myometrial contractions. Thickening or disruption of the layer by intramural fibroids may also contribute to a poor reproductive outcome, including infertility or early pregnancy loss [30,32]. In contrast to the rest of the myometrium, the junctional zone changes under the influence of estrogen and progesterone. During the window of implantation, at about 5–7 days after ovulation, myometrial contractions are limited to a minimum; decidualization of the endometrium and the junctional zone occurs. Uterine natural killer cells (uNK) and macrophages are responsible for the differentiation of tissue during decidualization. uNK cells are the most abundant and important immune cells in the uterus at the time of implantation. An alteration of uNK cell numbers has been associated with implantation failure [32,33].

The presence of fibroids appears to influence the number of uNK cells and macrophage cells. Kitaya et al. analyzed those cells in samples obtained after hysterectomy; the authors compared cell counts near fibroids with cells on the contralateral side of the uterus, far away from fibroids. In the mid-secretory phase, uNK cells were significantly reduced and macrophage cells significantly increased in the endometrium near fibroids compared to endometrium away from the fibroids, and also significantly reduced compared to healthy controls [34]. Regrettably, the study provides no data about the location of the fibroids. Furthermore, the mean age of women with fibroids, as well as of the healthy controls, was 40 years. They were candidates for hysterectomy, but not representative of the typical patient population suffering from infertility and recurrent pregnancy loss.

A physical disruption of the junctional zone, caused by intramural fibroids, may also lead to implantation failure or early pregnancy loss [35]. The expression of estrogen and progesterone, as well as their receptors, was reported to be altered at the junctional zone. However, this aspect needs further investigation [36,37].

### Uterine Myometrial Peristalsis

Cine-mode magnetic resonance imaging (MRI) permits analysis of myometrial contractions in the uterus [38]. The frequency of contractions appears to increase from menses to the mid-ovulatory phase of the cycle, and the contractions progress from the cervix to the fundus. The frequency is reduced after ovulation and especially during the time of implantation. The direction of peristalsis is also reversed in the luteal phase [39]. Compared to healthy controls, women with intramural and submucosal fibroids had increased myometrial peristalsis during the mid-luteal phase and decreased peristalsis in the peri-ovulatory phase [40,41]. Fifteen patients with intramural fibroids and a high frequency of uterine peristalsis in the mid-luteal phase were followed in a retrospective study. After myomectomy, peristalsis returned to normal in 14 of 15 patients; a pregnancy rate in excess of 40% was observed in the course of one year after surgery [42].

Leiomyomas are surrounded by a fibroid pseudocapsule (PC) that can be best identified during surgery, at the time of myomectomy. It consists of a bundle of smooth muscle cells and a vascular capsule responsible for blood supply. The PC is rich in neurotransmitters and neurovascularization. Endoglin and CD34, markers of neovascularization, are upregulated in the PC compared to the fibroid itself and the surrounding myometrium. The thickness of the capsule varies according to fibroid type and location. Submucosal fibroid PCs are significantly thicker than intramural myoma PCs, and intramural PCs are significantly thicker than subserosal PCs. The thickness also increases when the fibroid is located closer to the cervix [30,43]. The latter PCs are marked by higher expressions of enkephalin and oxytocin. These neuropeptides may alter fertility by inducing abnormal uterine contractions [44]. Furthermore, the intramural fibroid PC has been associated with increased levels of neurotensin, neuropeptide tyrosine, and the protein gene product 9.5 [44], all of which may induce muscular contractions. Large intramural fibroids might cause premature uterine contractions and disrupt early pregnancies, or cause preterm delivery [30,45].

## 4. Diagnosis

Ultrasonography, preferably by the transvaginal route, is the first-line diagnostic imaging procedure for the detection of fibroids. It is a widely available, economical, non-invasive and painless means of investigating the uterine cavity. Ultrasound is known for its high sensitivity and specificity in identifying fibroids. The size, exact location, and potential presence of fibroids in the uterine cavity can be assessed. After infusion of saline into the uterine cavity, transvaginal ultrasound is able to demonstrate submucosal fibroids and indicate the proximity of intramural fibroids to the cavity [46]. Figure 3a,b show a fibroid on 2D ultrasound and hysterosonography.

A “normal” 2D transvaginal ultrasound may be supplemented with a 3D transvaginal ultrasound. The latter permits reconstruction of the coronal plane of the uterus and thus demonstrates the exact location of the fibroid and distortion of the cavity due to submucosal fibroids [47,48]. Figure 4a,b show a myoma on 2D and 3D vaginal ultrasound.

On ultrasound examination, a uterine fibroid is typically seen as a well-defined round lesion within the myometrium or belonging to it, frequently with shadows at the edge or an internal fan-shaped shadow [49]. Doppler ultrasound reveals circumferential flow around the fibroid, called “ring of fire” [35]. Fibroids are usually hypoechoic or isoechoic. The echogenicity varies, depending on the level of calcification and the quantity of fibrous tissue. Sometimes a fibroid has anechoic components due to advancing necrosis. The size of the fibroid is estimated by measuring its three largest orthogonal diameters. Additionally, the minimum distance from the fibroid to the serosal surface and the endometrium of the uterus is measured [49].

The differential diagnosis of uterine masses is of crucial importance. Adenomyosis, endometrial polyps, or solid tumors of the adnexa are some of the most common misdiagnosed pathologies. Adenomyosis may be difficult to diagnose. A distinction is made between diffuse and focal adenomyosis, which are differentiated from adenomyomas. On histological investigation, adenomyomas are marked by additional compensatory hypertrophy of the surrounding myometrium [49]. Differentiating this condition from myoma can be challenging, especially when both pathologies are present together. Color Doppler ultrasound may be useful in this setting. Ultrasound findings that indicate the presence of adenomyosis include an asymmetrical thickening of the wall, so-called striae-like vascular patterns, fan-shaped shadowing, myometrial cysts, hyperechoic islands, echogenic buds and strips, and an irregular or interrupted junctional zone [49].

In cases of ambiguous ultrasound findings, magnetic resonance imaging (MRI) provides additional information (specificity 100%, accuracy 97%, and sensitivity 86–92%) [50]. Furthermore, in cases of huge fibroids, or especially in cases of multiple fibroids, when the shadow makes evaluation impossible, there is an indication for MRI.

A hysteroscopy should be performed for an even more detailed investigation or to confirm the potential involvement of the uterine cavity. During hysteroscopy the gynecologist may perform an endoscopy of the uterine cavity without anesthesia, usually even without hooking the cervix. The small optical instrument measuring just 3 mm in diameter serves the purpose of inspection. Figure 5 shows the hysteroscopic view of an inconspicuous uterine cavity with a raised endometrium in the center.

## 5. Management

Treatment options for fibroids include surgery, medication, and interventional radiology. The treatment improves symptoms by reducing the size of the fibroids, controlling abnormal uterine bleeding, or even curing the fibroids [51].

The key question is: When should the clinician treat a fibroid in women with infertility? It primarily depends on the existing clinical symptoms, as well as the size and location of the fibroids. The indications for treatment should be established with care because the association between infertility and fibroids may not be evident in some situations. Indications for surgery in intramural fibroids should be evaluated very carefully because surgery involves removal of the fibroid, but also causes scarring of the uterus wall, which may affect subsequent pregnancies. Medication may be used to treat abnormal uterine bleeding, although this approach has no more than a transient effect on fibroids. Available medical treatments include gonadotropin-releasing hormone (GnRH) agonists or antagonists, anti-progestins, progesterone-only treatments, combined hormonal contraceptives, selective progesterone receptor modulators (SPRMs), anti-fibrinolytic agents, and non-steroidal anti-inflammatory drugs (NSAIDs) [51]. In certain cases, GnRH agonists may be used before surgery in order to shrink fibroids and restore hemoglobin levels in symptomatic patients. However, due to their side effects, GnRH agonists cannot be used for a long time [51].

Nevertheless, in cases of fibroids and infertility, surgical therapy is to be given preference over medicinal therapy. Nevertheless, this therapy option should not remain unmentioned.

A thorough preoperative assessment is essential to determine the surgical strategy according to the size, location, and number of fibroids. A precise preoperative diagnosis will indicate whether a hysteroscopic resection or a laparoscopic myomectomy is feasible, and whether a laparotomy should be performed for numerous or large fibroids [52,53,54]. Each approach has its own indications. Currently, hysteroscopic myomectomy is the gold standard for surgical treatment of submucosal fibroids (FIGO 0 and 1 fibroids) [55,56]. FIGO 2 fibroids are more difficult to resect and may require a two-stage treatment, especially if they are larger than 3 cm in size [52].

Complications during the intervention are rare and mainly related to the difficulty of the surgical procedure. The most common problems associated with hysteroscopic myomectomy include uterine perforation, bleeding, infection, and venous intravasation [57,58]. Long-term complications such as intrauterine adhesions were reported in about 10% of cases during second-look hysteroscopy; the risk is higher in cases of multiple apposing fibroids [59]. Prevention strategies include the insertion of a postoperative intrauterine device (IUD), intrauterine balloons, hyaluronic acid gel, or postoperative treatment with oral estrogens to stimulate endometrial regeneration [59]. Surgical strategies may also permit prevention of adhesions. Monopolar resectoscopes appear to increase the risk of postoperative intrauterine adhesions compared to bipolar resection of fibroids [60]. However, evidence regarding prevention strategies is very limited. The duration of endometrial wound recovery varies for the different types of hysteroscopic surgery, ranging from one month after polypectomy to three months after myomectomy. The duration of wound recovery is important for subsequent fertility treatments [61].

Intramural and subserosal fibroids (FIGO 3 fibroids and above) are best removed by laparoscopy or laparotomy. Laparoscopic surgery is the first choice in the absence of contraindications. Laparoscopic myomectomy is considered more difficult by many gynecological surgeons, but its benefits are noteworthy: less postoperative pain, shorter hospital stays, less blood loss, and faster recovery. No difference was registered between the laparoscopic and abdominal approach in regard of reproductive outcomes [62]. Challenges in surgery include the appropriate use of sutures and the achievement of satisfactory hemostasis. The most frequent intraoperative complications of laparoscopic myomectomy include myometrial hematoma, excessive blood loss, and morcellation accidents [63,64].

From the fertility point of view there is a crucial importance to the dissection plane (the fibroid pseudocapsule) during surgical approach (both hysteroscopy and abdominal-laparoscopy/laparotomy). Several studies have analyzed the fibroid pseudocapsule, a neurovascular bundle surrounding uterine fibroids, which separates the fibroid from normal tissue [65,66]. This structure is well-known and can be histologically and sonographically examined. Tinelli et al. investigated the thickness of the pseudocapsule according to the uterine location of the fibroid. Compared to intramural and subserous fibroids they show that the capsule was thicker near the endometrial cavity, concluding a relevancy for myometrial healing and fertility. During myomectomy they recommended a systematic preservation of the pseudocapsule [65,66].

Complex conditions would be the presence of concomitant pathologies such as adenomyosis or adenomyoma, or the need for large intramural fibroid extraction [64]. Anti-adhesive agents may be useful in reducing postoperative adhesions [63]. Obstetric complications during labor are caused mainly by a weak myometrium after destruction due to extensive coagulation, defective suturing, and poor tissue approximation. The rate of uterine rupture in a subsequent pregnancy is reported to be 1% [64]. During laparoscopic myomectomy, fibroids are usually removed with a morcellator. Although the prevalence of leiomyosarcoma is very rare in fibroids (<0.3%), the risk of uterine fragment dispersion during morcellation remains a highly debated issue and has been addressed by many international societies [51,62].

Contraindications to laparoscopic myomectomy include multiple fibroids (>4) at different sites of the uterus, requiring numerous incisions, and the presence of an intramural fibroid >10–12 cm in size or suspected of being a leiomyosarcoma [62].

Figure 6 shows laparoscopic enucleation of a fibroid with reconstruction of the uterine wall.

High intensity focused ultrasound (HIFU) is a relatively recent noninvasive tumor ablative technique. With the aid of special transducers, the ultrasound waves are focused on an area measuring just a few millimeters. In the target organ, this leads to high temperatures, resulting in necrosis in parts of the fibroid. Over the last two decades, HIFU has been widely used in practice to treat uterine fibroids. As a noninvasive treatment technique, HIFU can be performed under the guidance of ultrasound (US-HIFU) or magnetic resonance imaging (MR-HIFU), so that the fibroids can be selectively ablated without damaging adjacent structures. This method requires minimal hospitalization, has no surgical wound, and has good relief and outcomes in many patients [67,68]. HIFU appears to be a safe and effective treatment in patients with uterine fibroids who wish to conceive and shows equivalent outcome in pregnancy rates compared to laparoscopic fibroid resection. [68,69].

Another treatment option is the radiofrequency ablation (RFA) as a form of hyperthermic ablation, using elevated temperature to produce tissue destruction. RFA of fibroids can be performed as a laparoscopic procedure. The RFA handpiece is directed into each myoma with simultaneously laparoscopic and ultrasound guidance. Ultrasound is utilized to verify appropriate placement of the device within each fibroid [70]. Alternatively, transvaginal RFA can be performed.

A study of the current literature shows that most of the papers on RFA analyze the outcomes concerning women with symptomatic fibroids. For example, the effect on bleeding disorders and dysmenorrhea, quality of life, or adverse events is primarily studied [71]. The relationship between RFA and infertility is rarely the focus [72]. However, individual papers address case reports comparing pregnancy rates after RIF with those after surgical treatment and find similar outcomes concerning pregnancy, live birth, and abortion rates [70,73].

In view of the absence of long-term data concerning fibroids and infertility, uterine artery embolization is not recommended [62,74].

## 6. Recommendation

Clinical pregnancy rates were high after myomectomy in patients with submucosal fibroids, but the ongoing pregnancy/live birth rate did not reach statistical significance. No change was registered in abortion rates [4,42].

Subserosal fibroids do not seem to affect fertility outcomes, and removal does not confer benefit [6,19].

In contrast to submucosal fibroids, recommendations concerning intramural fibroids that cause no distortion of the uterine cavity are far from clear. There is no consensus as to whether intramural fibroids should be removed in women with infertility. Many clinicians would recommend removal of intramural fibroids if they are ≥5 cm in diameter. A study performed by Hart et al. showed lower implantation/pregnancy and ongoing pregnancy rates in women with large (≥5 cm) intramural fibroids [75]; the authors recommend myomectomy in these cases. The procedure should be discussed individually with each patient, taking other potential conditions such as dysmenorrhea or irregular bleeding into account.

Some authors have registered no clear benefits for surgery and do not recommend the approach. However, the limitation of these studies is that they provide no clear information about the size, number, and location of fibroids. Although intramural fibroids are reported to be associated with poorer pregnancy outcomes, women who underwent myomectomy for intramural fibroids experienced no benefit with regard to pregnancy outcomes compared to controls. Regrettably, studies addressing this specific question and included in the Cochrane analysis are scarce and do not provide precise recommendations [19,76].

## 7. Conclusions

Pregnancy and live birth rates appear to be reduced in women with submucosal fibroids. Resection of these fibroids improves pregnancy rates. In contrast, subserosal fibroids do not affect fertility outcomes, and their removal does not confer any benefit. Intramural fibroids appear to reduce fertility, but recommendations concerning their treatment remain ambiguous. Myomectomy should be discussed individually with the patient. In addition to the problem of infertility, potential symptoms such as dysmenorrhea or bleeding disorders should be evaluated and included in the indication for surgery. A conclusive analysis of the value of myomectomy for the treatment of intramural fibroids requires further studies with due attention to the size and number of fibroids, as well as their distance to the endometrium.

a.Submucosal fibroids should be removed before ART or in cases of habitual abortions.b.Subserosal fibroids: as they do not seem to affect pregnancy rates, myomectomy does not appear to be necessary.c.Intramural fibroids: controversial data, lack of homogenous opinion. Intramural fibroids ≥5 cm: perform surgery before ART or in cases of habitual abortion. Intramural fibroids <5 cm: the reported outcome varies between no difference and significantly reduced cumulative pregnancy rates.

## Figures and Tables

**Figure 1 diagnostics-11-01455-f001:**
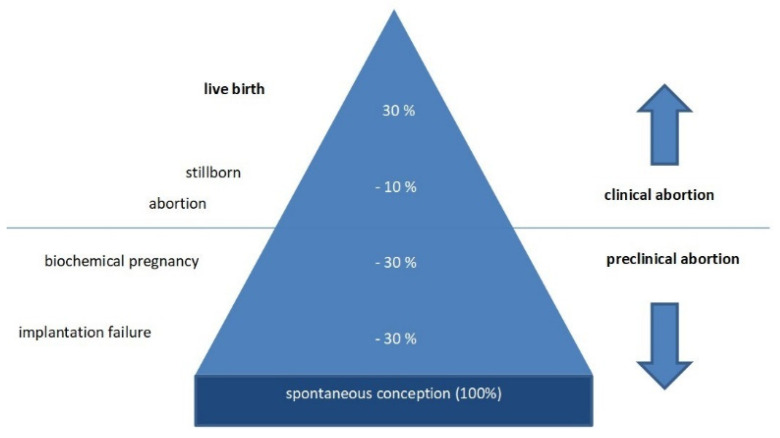
Human reproduction is an inefficient process. Only about 30% of conceptions (100%) result in a live birth. (modified according to [3]).

**Figure 2 diagnostics-11-01455-f002:**
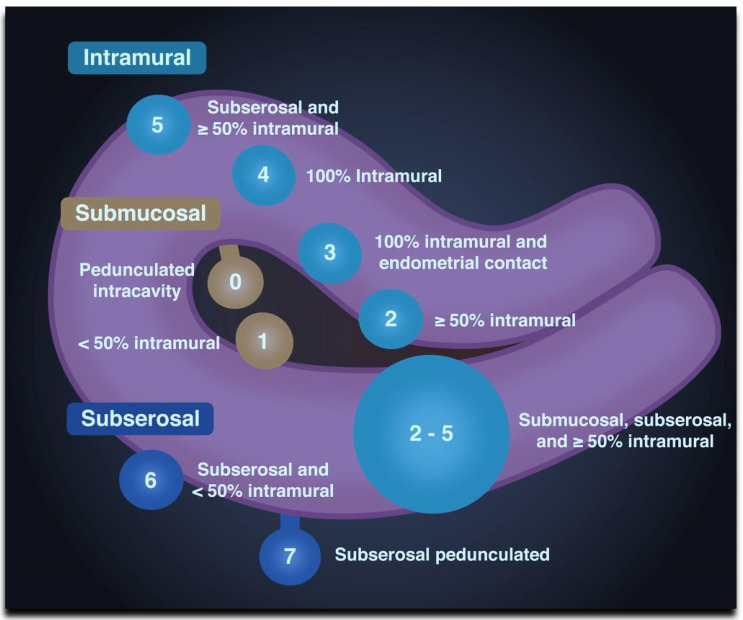
The International Federation of Gynecology and Obstetrics (FIGO) uses the above-depicted classification system of fibroids, organized according to the location between the submucosal, intramural, and subserosal layers of the uterus (modified according to Dr. Sachintha Hapugoda, Radiopaedia, Australia).

**Figure 3 diagnostics-11-01455-f003:**
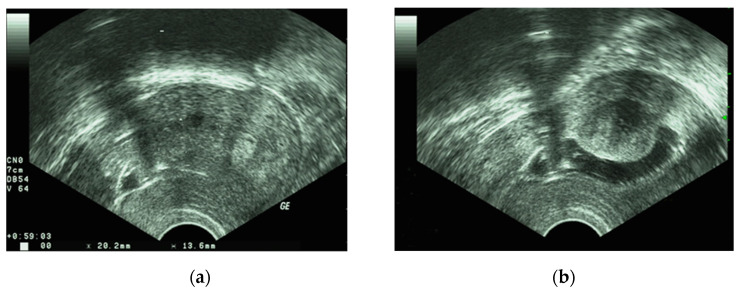
(**a**) fibroid on 2D ultrasound; (**b**) fibroid on hysterosonography.

**Figure 4 diagnostics-11-01455-f004:**
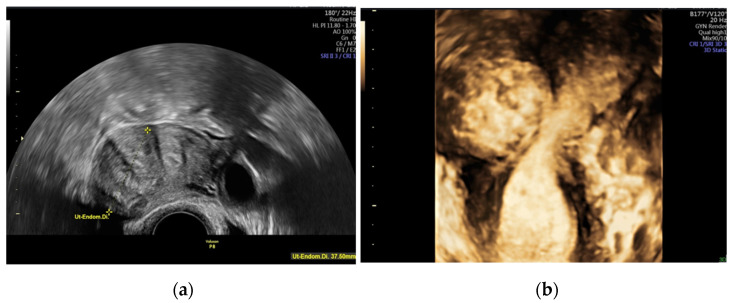
Presentation of an intramural myoma, affecting the cavum uteri, (**a**) with regular 2D vaginal ultrasound on the left-hand side and (**b**) with 3D vaginal ultrasound on the right-hand side.

**Figure 5 diagnostics-11-01455-f005:**
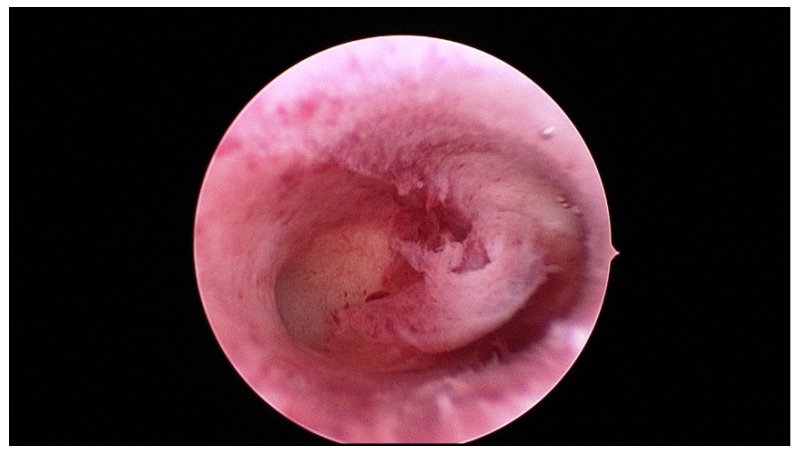
Hysteroscopic view of an inconspicuous cavum uteri with raised endometrium in the middle.

**Figure 6 diagnostics-11-01455-f006:**
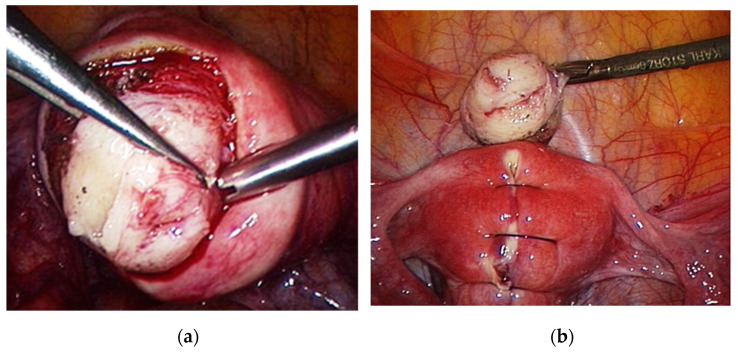
(**a**) laparoscopic enucleation of a fibroid with (**b**) reconstruction of the uterine wall.

**Table 1 diagnostics-11-01455-t001:** The International Federation of Gynecology and Obstetrics (FIGO) classification of fibroids.

	Type	Location
Submucosal	0	Pedunculated intracavitary
	1	<50% intramural
Intramural	2	≥50% intramural
	3	Contact with the endometrium, 100% intramural
	4	
	5	Intramural Subserosal ≥50% intramural
Subserosal	6	Subserosal <50% intramural
	7	Subserosal pedunculated
	8	Other (e.g., cervical, intraligamentous).
Hybrid (having contact with both the endometrium and the serosal layer). The numbers are listed separately with a hyphen. The first refers to the relationship with the endometrium, and the second refers to the relationship with the serosa.	2–5	Submucosal and subserosal, each with less than half the diameter in the endometrial and peritoneal cavities, respectively.

## Data Availability

Not applicable.

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
