# Peer review of "Uterine Fibroids and Infertility"

_diagnostics, 2021, doi:10.3390/diagnostics11081455_

Round 1
Reviewer 1 Report
The role played by uterine fibroids in women's fertility is a matter of controversy that has been insufficiently investigated. In their review Freytag et al. try to summarize the current information about this topic but this manuscript, although well written, adds few knowledge to there current body of literature and has several pitfalls that preclude its publication:
-The introduction section is too long with redundant information and and can easily be shortened.
-The manuscript contains several controversial affirmations, for example about the role of MRI of about the gold standard for evaluating the uterine cavity in the setting or fibroids, the role and effect of some medications on fibroids or fertility in this context.
-Unfortunately, the manuscript does not address the role of new less invasive treatments like radio frequency or HIFU
The bibliographic references are old (only 14/65 are from 2015 to now and only 7 are within the last 5 years)
Author Response
Specific comment of Reviewer #1:
- The introduction section is too long with redundant information and can easily be shortened.
We thank the reviewer for this important note and shortened the introduction accordingly.
- The manuscript contains several controversial affirmations, for example about the role of MRI of about the gold standard for evaluating the uterine cavity in the setting or fibroids, the role and effect of some medications on fibroids or fertility in this context.
We thank the reviewer for this important comment. Concerning the MRI, we mentioned the use of MRI in a study in order to analyze the frequency of contractions. Furthermore, we added “In cases of ambiguous ultrasound findings, magnetic resonance imaging (MRI) provides additional information (specificity 100%, accuracy 97%, and sensitivity 86-92%) [53].” Here, we do not agree with the criticism concerning controversial affirmations about the role of MRI and the gold standard for evaluating the uterine cavity.
We reformulated the effect of some medications on fibroids and fertility.
- Unfortunately, the manuscript does not address the role of new less invasive treatments like radio frequency or HIFU.
We thank the reviewer for this important note and added a paragraph concerning radiofrequency ablation and HIFU.
- The bibliographic references are old (only 14/65 are from 2015 to now and only 7 are within the last 5 years)
The reviewer is absolutely right with this point of criticism. We have conducted another literature search and added a few current references to the manuscript.
Reviewer 2 Report
1. During the introduction, I believe that the relation between Vit D and fibroids should be mentioned
(Lima et al. Reproductive Biology and Endocrinology (2021) 19:67 https://doi.org/10.1186/s12958-021-00752-x
https://doi.org/10.1016/j.fertnstert.2021.02.040
Vergara, D.; Catherino, W.H.; Trojano, G.; Tinelli, A. Vitamin D: Mechanism of Action and Biological Effects in Uterine Fibroids. Nutrients 2021, 13, 597. https://doi.org/10.3390/nu13020597
https://doi.org/10.1016/j.fertnstert.2020.09.151
Sheng B, Song Y, Liu Y, et al. Association between vitamin D and uterine
fibroids: a study protocol of an open-label, randomised controlled trial. BMJ Open 2020;10:e038709. doi:10.1136/bmjopen-2020-038709)
2. When you state that SM fibroids can be divided (line 142: Submucosal fibroids (with and without distortion of the cavity), by definition a SM fibroid is distorting the cavity by being inside or growing from the myometrium into the cavity (Type 2, less than 50% inside the cavity). I do not think that it should be divided.
3. Lines 268-9, agree that the use of MRI is when US is ambiguous, but usually when you find a big myoma or especially in the case of multiple myomas where the shadow will not allow you to see, there is an indication for MRI
4. Line 250 - Doppler ultrasound reveals circumferential flow 250 around the fibroid: the name, due to the doppler image, is "ring of fire"
Andrea Tinelli et al. A Combined Ultrasound and Histologic Approach for Analysis of Uterine Fibroid Pseudocapsule Thickness. Reproductive Sciences 2014, Vol. 21(9) 1177-1186
5. During surgical approach (both hysteroscopy and abdominal - laparoscopy/laparotomy) there is a crucial importance to the dissection plane (the pseudocapsule) form the fertility point of view. It is impossible today to talk about reproductive surgery of fibroids without bringing this fact. I would like the author to focus more on this aspect.
A. Tinelli, et al., The importance of pseudocapsule preservation during hysteroscopic myomectomy, Eur J Obstet Gynecol (2019), https://doi.org/10.1016/j.ejogrb.2019.09.008
Andrea Tinelli et al., Submucous Fibroids, Fertility, and Possible Correlation to Pseudocapsule Thickness in Reproductive Surgery. BioMed Research International Volume 2018, Article ID 2804830, 7 pages https://doi.org/10.1155/2018/2804830
Cochrane review on SM myoma and subfertility is based only in one paper, there is much more evidence to support surgical approach in cvase of infertility.
Author Response
Specific comment of Reviewer #2:
- During the introduction, I believe that the relation between Vit D and fibroids should be mentioned.
- (Lima et al. Reproductive Biology and Endocrinology (2021) 19:67 https://doi.org/10.1186/s12958-021-00752-x
- https://doi.org/10.1016/j.fertnstert.2021.02.040
- Vergara, D.; Catherino, W.H.; Trojano, G.; Tinelli, A. Vitamin D: Mechanism of Action and Biological Effects in Uterine Fibroids. Nutrients 2021, 13, 597. https://doi.org/10.3390/nu13020597
- https://doi.org/10.1016/j.fertnstert.2020.09.151
- Sheng B, Song Y, Liu Y, et al. Association between vitamin D and uterine fibroids: a study protocol of an open-label, randomised controlled trial. BMJ Open 2020;10:e038709. doi:10.1136/bmjopen-2020-038709)
We thank the reviewer for this important note and added a paragraph concerning vitamin D and fibroids to the introduction.
- When you state that SM fibroids can be divided (line 142: Submucosal fibroids (with and without distortion of the cavity), by definition a SM fibroid is distorting the cavity by being inside or growing from the myometrium into the cavity (Type 2, less than 50% inside the cavity). I do not think that it should be divided.
We thank the reviewer for this note and removed this part.
- Lines 268-9, agree that the use of MRI is when US is ambiguous, but usually when you find a big myoma or especially in the case of multiple myomas where the shadow will not allow you to see, there is an indication for MRI.
The reviewer is right with this point of criticism. We added this aspect accordingly.
- Line 250 - Doppler ultrasound reveals circumferential flow 250 around the fibroid: the name, due to the doppler image, is "ring of fire".
Andrea Tinelli et al. A Combined Ultrasound and Histologic Approach for Analysis of Uterine Fibroid Pseudocapsule Thickness. Reproductive Sciences 2014, Vol. 21(9) 1177-1186
We thank the reviewer for this interesting note and added it accordingly.
- During surgical approach (both hysteroscopy and abdominal - laparoscopy/laparotomy) there is a crucial importance to the dissection plane (the pseudocapsule) form the fertility point of view. It is impossible today to talk about reproductive surgery of fibroids without bringing this fact. I would like the author to focus more on this aspect.
- Tinelli, et al., The importance of pseudocapsule preservation during hysteroscopic myomectomy, Eur J Obstet Gynecol (2019), https://doi.org/10.1016/j.ejogrb.2019.09.008
- Andrea Tinelli et al., Submucous Fibroids, Fertility, and Possible Correlation to Pseudocapsule Thickness in Reproductive Surgery. BioMed Research International Volume 2018, Article ID 2804830, 7 pages https://doi.org/10.1155/2018/2804830
We thank the reviewer for this interesting note and added this aspect.
- Cochrane review on SM myoma and subfertility is based only in one paper, there is much more evidence to support surgical approach in case of infertility.
We thank the reviewer for this interesting note and added it accordingly.
- Falcone, T., Parker, W.H., Surgical management of leiomyomas for fertility or uterine preservation. Obstet Gynecol, 2013. 121: p. 856–868.
- Pakrashi, T., New hysteroscopic techniques for submucosal uterine fibroids. Current Opinion in Obstetrics and Gynecology, 2014. 26(4): p. 308–313.
- Casadio, P., et al., Hysteroscopic myomectomy: Techniques and preoperative assessment. Minerva Ginecol, 68: p.154–166.
- Zepiridis, L.I., et al. Infertility and uterine fibroids. Best Practice Research Clinical Obstetrics Gynaecology, 34: p.66–73.